# Establishment and Use of Primary Cultured Astrocytes from Alexander Disease Model Mice

**DOI:** 10.3390/ijms252212100

**Published:** 2024-11-11

**Authors:** Yuto Kubota, Eiji Shigetomi, Kozo Saito, Youichi Shinozaki, Kenji Kobayashi, Masayoshi Tanaka, Bijay Parajuli, Kenji F. Tanaka, Schuichi Koizumi

**Affiliations:** 1Department of Neuropharmacology, Interdisciplinary Graduate School of Medicine, University of Yamanashi, Chuo 409-3898, Japan; g22dim02@yamanashi.ac.jp (Y.K.); eshigetomi@yamanashi.ac.jp (E.S.); ksaitoko@yamanashi.ac.jp (K.S.); shinozaki-yi@igakuken.or.jp (Y.S.); kenjikoba.yg@gmail.com (K.K.); tanakama@oita-u.ac.jp (M.T.); parajuli@yamanashi.ac.jp (B.P.); 2GLIA Center, University of Yamanashi, Chuo 409-3898, Japan; 3Division of Brain Sciences, Institute for Advanced Medical Research, Keio University School of Medicine, Shinjuku, Tokyo 160-8582, Japan; kftanaka@keio.jp

**Keywords:** cultured astrocytes, cell-autonomous effect, Rosenthal fibers (RFs)

## Abstract

Alexander disease (AxD) is an intractable neurodegenerative disease caused by mutations in *glial fibrillary acidic protein* (*GFAP*), which is predominantly expressed in astrocytes. Thus, AxD is a primary astrocyte disease. However, it remains unclear how *GFAP* mutations affect astrocytes and cause AxD pathology. Three features are characteristic of AxD astrocytes *in vivo*: (1) Rosenthal fibers (RFs), the hallmark of AxD; (2) aberrant Ca^2+^ signals (AxCa); and (3) upregulation of disease-associated genes (AxGen). We established a primary culture system for astrocytes from an AxD transgenic mouse model, and used it to analyze the above features of AxD pathogenesis in astrocytes *in vitro*. We observed the formation of RFs in AxD primary cultures. The abundance of RFs was greater in AxD-transgene-homozygous compared with -hemizygous astrocytes, indicating a gene dosage effect, and this abundance increased with time in culture, indicating a developmental process effect. However, cultured AxD astrocytes did not exhibit changes in either AxCa or AxGen. We therefore conclude that RFs in astrocytes form via a cell-autonomous mechanism, whereas AxCa and AxGen are likely to occur via a non-cell-autonomous mechanism through interactions with other cells, such as neurons, microglia, and vascular cells. Although primary cultured AxD astrocytes are suitable for elucidating the mechanisms of RFs formation and for intervention studies, it should be noted that they cannot reflect the pathophysiology of non-cell-autonomous events in astrocytes.

## 1. Introduction

Alexander disease (AxD) was first reported in 1949 by Dr. Alexander in an infant with mental retardation, convulsions, and progressive hydrocephalus [1]. The clinical phenotypes of AxD are varied and include macrocephaly, frontal leukodystrophy, and a variety of developmental delays with epileptic seizures, an inability to ambulate because of muscle weakness in the extremities, dysphagia, and psychomotor developmental delay [2,3,4,5]. MRI is used to assist in the diagnosis of AxD, which includes extensive white matter changes in the frontal predominance [6]. However, AxD is a very rare neurodegenerative disease; therefore, it has received limited attention since its initial description. In 2001, Dr. Brenner and colleagues demonstrated that it was caused by missense mutations in *glial fibrillary acidic protein* (*GFAP*), a gene that is predominantly expressed in astrocytes [7]. Since this discovery, significant advances have been made in AxD diagnosis and in understanding the disease’s pathology. For example, AxD can now be genetically diagnosed, effects of *GFAP* mutants on cell biology have been investigated [8,9,10,11], and several genetically engineered mice with *GFAP* mutations such as R79H [12] and R239H [13] have been created to advance our understanding of AxD. However, it remains unclear how mutations in *GFAP* cause the neurological symptoms of AxD described above. Furthermore, effective therapies based on the underlying molecular pathogenesis have not been developed.

We generated an AxD model mouse overexpressing human *GFAP* carrying the AxD mutation, R239H [13]. Astrocytes in these AxD mice have the following three biochemical, functional, and genetic features: (1) aggregation of Rosenthal fibers (RFs), (2) aberrant extra-large Ca^2+^ signals (AxCa), and (3) upregulation of genes characteristic of AxD (AxGen) [14]. As AxD is a primary astrocyte disease, it is postulated that severe neurological deficits are triggered by astrocyte dysfunction. The astrocyte abnormalities observed in our AxD mice are therefore likely to be involved in the molecular pathogenesis of AxD.

To investigate these phenotypes further, we developed a primary culture system for astrocytes derived from the *GFAP* R239H AxD model mouse to enable their properties to be analyzed *in vitro*. We aimed to answer the following two questions. 1. Do the properties of cultured primary AxD astrocytes reflect the properties of astrocytes in the brain? It is well known that the properties of glial cells differ greatly between *in vitro* and *in vivo* states, and that *in vitro* data cannot be reliably extrapolated to the *in vivo* state [15,16]. We therefore first clarify the usefulness and limitations of AxD astrocyte primary cultures by comparing the three representative features of *in vivo*/*in situ* AxD astrocytes. 2. Are the *in vivo* characteristics of AxD astrocytes cell-autonomous? Although AxD is a congenital genetic disorder, it sometimes develops or worsens suddenly, perhaps in response to infection or a traumatic brain injury [17,18]. Therefore, it is highly possible that astrocyte properties are regulated by interactions with other cells, especially immune cells. Using *GFAP* R239H AxD mice, we recently reported that AxD pathogenesis is regulated by microglia, resident innate immune cells of the central nervous system [19]. A culture system of only AxD astrocytes eliminates non-cell-autonomous regulation of AxD astrocytes by other cells, such as microglia, neurons, and vascular cells, and simplifies the analysis of AxD astrocytes.

In this study, we established primary cultures of AxD astrocytes. We used these cultures to demonstrate that the mechanism of RFs formation, a hallmark of AxD, is cell-autonomous, whereas AxCa and AxGen are caused by non-cell-autonomous mechanisms. These findings suggest that cultured AxD astrocytes are useful for the analysis of RFs, but not for that of AxCa and AxGen.

## 2. Results

### 2.1. RFs Are Formed in Primary Cultured AxD Astrocytes

RFs in astrocytes are a common pathological feature in the brains of AxD patients and in animal models of AxD. They are intracellular inclusions of aggregated cytoskeletal proteins, such as GFAP and vimentin, and heat shock proteins [12,13,20,21,22,23,24]. We first cultured primary astrocytes from transgenic mice overexpressing mutant human *GFAP* R239H to investigate whether cultured AxD astrocytes recapitulate RFs. RFs in brain tissue can be stained by the fluorescent dye, Fluoro-Jade B (FJB) [25]; therefore, we used this stain to detect RFs *in vitro*. Perinuclear GFAP aggregates with honeycomb-like structures were observed in cultured AxD astrocytes, which were stained with FJB (Figure 1A). Such aggregates were completely absent in wild-type (WT) astrocytes. αB-crystallin, a kind of small heat shock protein, colocalized with GFAP aggregates (Figure 1B). Vimentin also colocalized with GFAP aggregates in vimentin^+^ GFAP^+^ astrocytes, although some astrocytes were stained for vimentin but not for GFAP (in WT and AxD cultures) (Figure 1C). The data clearly indicate that these inclusions are RFs.

### 2.2. RFs Abundance Depends on the Amount of Mutant GFAP

To examine the effects of mutant *GFAP* on RFs formation, we compared primary cultures of astrocytes from *GFAP* R239H AxD mice. Primary astrocyte cultures were prepared from mice that were hemizygous (with an estimated three transgene copies) or homozygous (with an estimated six transgene copies) for the *GFAP* R239H transgene. The percentage of FJB^+^ astrocytes in homozygotes was more than in hemizygotes (2.05% in hemizygotes, 19.5% in homozygotes) (Figure 2A,B). These percentages were similar to that of FJB^+^ cortical astrocytes observed in AxD model mice. These findings indicate that the formation of RFs is contingent upon the quantity of mutant GFAP.

### 2.3. The Number of RFs Increases with Culture Time

The number of RFs in *GFAP* R239H AxD mice increases with age [13]. To explore the mechanism of RFs formation, we quantified the number of FJB^+^ primary cultured AxD astrocytes (homozygous) on culture days (D) 7, 11, and 17. The number of both astrocytes and FJB^+^ astrocytes increased with days of culture (Figure 3A–C). Of note, the percentage of FJB^+^ astrocytes increased with increased culture time (D7: 16.7%; D11: 28.2%: D17: 41.2%) (Figure 3D). These findings indicate that RFs formation requires cell maturation and is likely regulated by cell-autonomous mechanisms, because cultured AxD astrocytes were pure astrocyte cultures.

### 2.4. Cultured AxD Astrocytes Do Not Exhibit AxCa

We next tested whether functional abnormalities occur in cultured AxD astrocytes, and whether they are caused by cell-autonomous mechanisms. Ca^2+^ signaling is a good indicator of astrocyte functions [26,27,28,29]. We previously performed *in situ* Ca^2+^ imaging of astrocytes in AxD model mice. We observed an abnormally large Ca^2+^ signal with high frequency, which contributes to the pathogenesis and disease progression of AxD. We termed this AxCa [14]. We therefore conducted *in vitro* Ca^2+^ imaging using Fura-2 to investigate AxCa in primary AxD astrocyte cultures. We analyzed the frequency and the peak amplitude of spontaneous Ca^2+^ oscillations for 10 min. However, there were no significant differences in amplitude or frequency of Ca^2+^ oscillations between WT and AxD astrocytes, i.e., cultured AxD astrocytes did not exhibit AxCa (Figure 4A–D). These findings indicate that AxCa observed *in situ* is not replicated in cultured AxD astrocytes.

### 2.5. Cultured AxD Astrocytes Exhibit Little Up-Regulation of AxGen

We previously identified genes that are upregulated in purified astrocytes derived from AxD mice using RNA-sequencing analysis. The most highly upregulated genes were *Lcn2*, *C3*, *Glycam1*, *Gdf15*, and *Gfap* [14]. We term these disease-associated genes as AxGen. Comparison of AxGen expression levels in cultured AxD astrocytes revealed endogenous mouse *Gfap* and *Lcn2* to be upregulated in a genotype-dependent manner (*Gfap*: 1.3-fold in hemizygotes, 1.8-fold in homozygotes; *Lcn2*: 1.8-fold in hemizygotes, 2.6-fold in homozygotes) (Figure 5A,B). However, this upregulation was much smaller compared with that *in vivo* in AxD mice (*Gfap*: 2.0-fold in hemizygotes, 5.0-fold in homozygotes; *Lcn2*: 35-fold in hemizygotes, 362-fold in homozygotes). We tested other AxGen and found little change in their expression levels in cultured AxD astrocytes (*C3*: 1.9 ± 0.1-fold in homozygotes; *Atp2a2*: 1.0 ± 0.06-fold in homozygotes). The increased expression of AxGen in astrocytes *in vivo* in AxD mice was less evident in cultured AxD astrocytes. Therefore, AxGen is not suitable for use with the culture system for analyzing the molecular pathogenesis of AxD astrocytes or for evaluating the action of drugs.

## 3. Discussion

Our findings in this study are as follows. (1) We observed RFs formation in AxD primary astrocyte cultures. RFs were more abundant in homozygous than in hemizygous *GFAP* R239H AxD model cells, indicating an effect of gene dosage. In addition, the number of astrocytes with RFs increased with the days in culture. (2) Cultured AxD astrocytes did not show AxCa. (3) Cultured AxD astrocytes also did not show AxGen. We therefore conclude that RFs form via a cell-autonomous mechanism, whereas AxCa and AxGen occur in a non-cell-autonomous manner through interactions with other cells, such as neurons, microglia, and vascular cells. These results suggest that primary culture of AxD astrocytes can be useful for studying RFs, but not for studying non-cell-autonomous events in astrocytes such as AxCa and AxGen.

### 3.1. Formation of RFs Is a Non-Cell-Autonomous Mechanism of Astrocytes

RFs within astrocytes are the major hallmark of AxD. RFs contain GFAP, heat shock proteins and vimentin [21,22,30]. Definitive images of RFs can be obtained by electron microscopy [20,23]; however, they can also be detected as aggregates containing GFAP by immunostaining and by FJB staining [13,25]. RFs have been previously stained *in vivo* by FJB; however, in this study, we successfully stained RFs *in vitro* for the first time. RFs in cultured AxD astrocytes were detected as GFAP aggregates by immunocytochemistry and by puncta-like FJB^+^ signals (Figure 1 and Figure 2). We also observed αB-crystallin- (Figure 1B) and vimentin-positive signals in RFs (Figure 1C). It should be noted that such RFs were not observed in cultured WT astrocytes. Together, these findings indicate that GFAP aggregates in cultured AxD astrocytes are caused by *GFAP* mutation, and should therefore be RFs.

When mutant *GFAP* was transfected into non-astrocyte SW13^vim-^ cells, an adenocarcinoma cell line lacking vimentin, GFAP aggregates were formed [8,31]. In addition, astrocytes derived from induced pluripotent stem cells of three different AxD patients with different genotypes contained GFAP aggregates [11]. Mutations in *GFAP* can therefore cause GFAP aggregates even if they are expressed in non-astrocyte cells or in cells with different genetic backgrounds, suggesting that a *GFAP* mutation itself may be sufficient for RFs formation. However, RFs can also be observed in the brains of other central nervous system diseases, such as multiple sclerosis and ciliary astrocytoma or when non-mutant *GFAP* is highly overexpressed [32,33]. It therefore appears that *GFAP* mutations are not a prerequisite for RFs formation. Perhaps RFs are formed by a complex environment that includes multiple factors including *GFAP* mutation.

In our AxD model mice, the number of FJB^+^ signals *in vivo* was higher in homozygous than in hemizygous mice. The present *in vitro* experiments also showed that the number of FJB^+^ astrocytes was greater when derived from homozygous compared with hemizygous *GFAP* R239H mice (Figure 2A,B). These *in vivo* and *in vitro* observations indicate that the formation of RFs is dependent on the amount of mutant *GFAP*. Indeed, increasing the ratio of purified mutant GFAP to WT GFAP promotes aggregate formation *in vitro* [31]. Mutant *GFAP* may therefore lower the threshold of RFs formation, and the amount of mutant *GFAP* may play an important role in the threshold.

In AxD model mice, the number of GFAP aggregates increases from postnatal day 7 to day 28 [13], and more RFs were formed at 9 weeks than at 6 weeks of age [14]. This suggests that RFs are formed in a developmentally time-dependent manner, at least in a certain time window. Similarly, an increase in the number of FJB^+^ astrocytes correlated with prolonged culture time (Figure 3), which was consistent with a previous report [34]. The fact that time-dependent RFs formation was also reproduced in pure astrocyte cultures indicates that this is caused by a cell-autonomous mechanism, such as astrocyte maturation.

Taken together, we demonstrated that formation of RFs in AxD astrocytes can be reproduced by this culture system, and is dependent on *GFAP* mutation, mutated *GFAP* gene dosage, and development. This *in vitro* culture system is expected to be useful for further studying the mechanism of RFs formation.

### 3.2. AxCa Was Not Observed in Cultured AxD Astrocytes

AxCa is a widespread spontaneous Ca^2+^ signal with large amplitude and high frequency that is observed in AxD astrocytes *in situ* [14]. AxCa is also a large intra-astrocyte Ca^2+^ signal, whose trigger is related to slight depletion of Ca^2+^ stores in the endoplasmic reticulum caused by mild reduction in the sarcoplasmic/endoplasmic reticulum Ca²⁺-ATPase, SERCA. Indeed, the SERCA2 gene *Atp2a2* was downregulated in AxD astrocytes *in vivo* [14]. In the present study, neither amplitude nor frequency of spontaneous Ca^2+^ signals were enhanced in cultured AxD astrocytes (Figure 4A–D), indicating that AxCa is not reproduced in cultured AxD astrocytes. In support of this, *Atp2a2* was not downregulated in AxD astrocytes *in vitro* (see Section 2). In contrast, aberrant Ca^2+^ signals have been reported in human-induced pluripotent stem cell-derived astrocytes, although they are partially distinct from AxCa: higher amplitude of ATP-induced Ca^2+^ signals caused by *ATP1b2* downregulation [35] and the deficits in Ca^2+^ propagation caused by reduced ATP release [9]. These results indicate that ATP-mediated Ca^2+^ signaling is different between cultured WT and AxD astrocytes. We previously revealed that AxCa is not dependent on ATP/P2 receptors as suramin, a P2 receptor antagonist, did not prevent AxCa [14]. Therefore, they are not necessarily inconsistent with the fact that our AxCa did not differ between cultured WT and AxD astrocytes in this study. Although further investigation is needed, a proper elucidation of the AxCa mechanism would be the most effective means of understanding why no AxCa-like Ca^2+^ signal was observed in cultured AxD astrocytes in the present study. AxCa is an *in situ* event; therefore, it is also important to determine whether interaction with other cells, such as neurons and microglia, is required for it to occur. In any case, the absence of AxCa in cultured AxD astrocytes clearly indicates that AxCa is not a suitable indicator for elucidating AxD pathogenesis using this culture system.

### 3.3. Expression of AxGen Was Minimally Affected in Cultured AxD Astrocytes

We previously performed microarray analysis of astrocytes derived from AxD mice and identified upregulated genes (e.g., *Lcn2*, *C3*, *Glycam1*, *Gdf15*, and *Gfap*) [14] that include reactive astrocyte markers [36,37]. In this study, we refer to these AxD-associated genes as AxGen. In our cultured AxD astrocytes, expression of AxGen, such as *Gfap*, *Lcn2*, and *C3*, was largely unaltered. In actuality, *Lcn2* and *C3* were slightly upregulated; however, their extent was much smaller than those *in vivo* (Figure 5A,B). The first reason why the expression of AxGen was not altered in the primary astrocyte culture system is that astrocytes are unable to communicate with other cells. Recently, we showed that in AxD model mice, microglia *in vivo* sense abnormalities in AxD astrocytes and significantly alter their properties when they contact astrocytes [19]. Therefore, the lack of significant changes in AxGen expression in AxD astrocytes is likely to be caused by a lack of interaction with multiple cell types, including microglia. The second reason is related to astrocyte maturity. It has been reported that astrocytes mature by long-term culture [38] or by interactions with neurons or endothelial cells [39]. Therefore, pure cultured astrocytes used in this study might remain immature. Whether long-time culture of astrocytes or co-cultures with other cells can reproduce AxGen will be the subject of our next study. Furthermore, in this study, we analyzed primary cultured astrocytes derived from both cortex and hippocampus for experiments. Since the heterogeneity of astrocytes among brain regions is well known [40,41], a third possibility is that the functional and molecular diversity of the astrocytes increased the variability of the data and that no AxGen was observed. Finally, properties of glial cells, such as morphology, molecular properties, and various responses and other functions, differ greatly between the culture system and the *in situ* state [15,16]. Therefore, a fourth reason for the failure to observe changes in AxGen expression is that changes caused by culturing astrocytes masked changes in AxGen expression. From these, we conclude that AxGen is not a suitable indicator for elucidating AxD pathogenesis using this culture system.

## 4. Materials and Methods

The reagents used in the various experiments are as follows (Table 1).

### 4.1. Mice

All animal care and experimental procedures were conducted in accordance with the “Guiding Principles in the Care and Use of Animals in the Field of Physiologic Sciences” published by the Physiologic Society of Japan and with prior approval of the Animal Care Committee of the University of Yamanashi (Chuo, Yamanshi, Japan). Pregnant wild-type mice (C57BL/6) at days 11–17 of gestation (E11–17) were purchased from SLC (Shizuoka, Japan). The AxD model mice were transgenic mice carrying multiple copies of a human *GFAP* cDNA with the mutation R239H, which is common in infant patients, under the control of a murine *GFAP* promoter [13]. The experiments were conducted using both hemizygous and homozygous genotypes of AxD model mice.

### 4.2. Primary Astrocyte Culture

Primary astrocyte culture was conducted by modifying a previous study [42]. On postnatal days 0–2 (P0–2), littermates were anesthetized with ice and the cerebral cortex and hippocampus were removed. The brain tissue was minced with a scalpel and collected in 0.025–0.2% trypsin-EDTA. DNase I was added at a concentration of 400 µg/mL and the mixture was shaken at 37 °C for 10 min at a speed of 100 times/min (personal-11, Taitec, Aichi, Japan). Cells were then suspended with a Pasteur pipette, and the enzyme reaction was terminated by the addition of culture medium (DMEM, EMEM, or DMEM/F12; 10% fetal bovine serum (FBS); 0.1% penicillin–streptomycin). The cell suspension was filtered through a 70 µm cell strainer and subjected to centrifugation (4 °C, 162× *g*, 10 min) (H-19FMR, Kokusan, Saitama, Japan). Subsequently, the supernatant was removed, and the cells were resuspended in culture medium. Cell suspensions derived from 3 brains were seeded into 75 cm^2^ flasks. Alternatively, 8-well chambers and 24-well plates were seeded at a concentration of 25 × 10^4^ cells/cm^2^. The cultures were incubated at 37 °C with the culture medium changed every 3 or 4 days.

### 4.3. RT-qPCR

After seeding, astrocytes were cultured in 24-well plates for 8 days. The cells were then washed 3 times with PBS (−), and RNA was extracted using the RNeasy Mini Kit (74106, QIAGEN, NW, Germany) according to the manufacturer’s instructions. Reverse transcription of the extracted RNA was performed using a PrimeScript RT Reagent Kit and a Verti 96 well Thermal Cycler (4375786, Applied Biosystems, MA, USA) at 37 °C for 15 min, 85 °C for 5 s, and 4 °C for an indefinite duration according to the manufacturers’ instructions. cDNA was added to 96-well plates containing IDT probe (Table 2), IDT Master Mix, and PCR was conducted using a Real-Time StepOnePlus instrument (Applied Biosystems). The data were analyzed using StepOne software, version 2.3, and the results were compared using the ΔΔCt method.

### 4.4. Immunocytochemistry

The antibodies used for immunocytochemistry and their corresponding dilution factors are presented in Table 3. The primary cultured astrocytes were washed 3 times with PBS (−) and subsequently fixed in 4% PFA for 15 min at 4 °C. Fixed cells were then incubated in blocking solution (0.1% Triton-X, 10% normal goat serum in PBS (−)) for 30 min. Primary antibody and 5% normal goat serum were then added and allowed to react for a minimum of 24 h at 4 °C. Following three 10 min washes in PBS (−), the secondary antibody was added and allowed to react for 1 h at room temperature. After three further washes in PBS (−), DAPI was added and allowed to react for 15 min. Finally, images were acquired using a confocal laser microscope (FV1200, OLYMPUS, Tokyo, Japan).

### 4.5. Fluoro-Jade B (FJB) Staining

To visualize RFs within astrocytes, FJB staining was performed in conjunction with immunostaining. After primary antibody reaction of at least 24 h, cells were treated with 100% ethanol for 3 min and then with 70% ethanol for 1 min. Cells were then incubated in a 0.06% potassium permanganate solution for 3 min. After washing with distilled water, a preconditioned FJB solution (0.0001% FJB, 0.09% acetic acid in distilled water) was added and allowed to stand for 5 min. After washing with distilled water, the secondary antibody and DAPI were added to complete the immunohistochemistry procedure.

### 4.6. In Vitro Ca^2+^ Imaging

The Ca^2+^ indicator, Fura2-AM, was employed to quantify changes in Ca²⁺ concentration in cultured astrocytes. Cultured cells, seeded in 75 cm^2^ flasks, were cultivated in astrocyte culture medium (DMEM or EMEM; 10% FBS; 0.1% penicillin–streptomycin) for 7 or 8 days. Thereafter, 0.1% trypsin-EDTA was used to detach adherent cells. Cell suspensions were prepared to a concentration of 20 × 10⁴ cells/mL and subsequently reseeded into 8-well chamber slides at a volume of 30 µL per well. On days 7–10 after reseeding, the cells were washed three times with a balanced salt solution (BSS). Fura2-AM (5 µM) was resuspended in BSS for 10 min using an ultrasonic cleaner (BRANSONIC CPX1800-J, Emerson Japan, Tokyo, Japan). The Fura2-AM solution was added at 245 µL/well and incubated for 60 min at room temperature. After Fura2-AM solution removal, cells were washed 3 times with BSS and allowed to stand for 60 min at room temperature in the absence of light. Observations were conducted using an inverted microscope (IX-73, Olympus) equipped with a 20× objective lens. Astrocytes loaded with Fura2-AM were alternately excited at 2 wavelengths, 340 nm and 380 nm, using a 75 W xenon lamp. The resulting fluorescence (F340 and F380) was acquired with a CCD camera (ORCA-R2, Hamamatsu Photonics, Shizuoka, Japan) installed in the microscope. The ratio of F340 to F380 (ΔF340/F380) was used as an indicator of Ca^2+^ concentration change. Microscope control and image acquisition were conducted using Aquacosmos software (Hamamatsu Photonics, Shizuoka, Japan). Observations were conducted for 10 min while cells were refluxed with BSS.

### 4.7. Image Analysis

Image data were subjected to analysis using the image processing software Fiji (https://imagej.net/software/fiji/, accessed on 7 November 2024).

### 4.8. Statistical Analysis

Numerical data were processed using OriginPro2020 (OriginLab, Newton, MA, USA) and Excel (Microsoft, Redmond, WA, USA). The Ca^2+^ imaging F340/F380 ratio was processed in Excel, and the waveforms were subsequently analyzed by peak analyzer in OriginPro2020. Numerical data, accompanied by graphical representations, are presented as the mean ± standard error (mean ± SEM). To ascertain whether there were any significant differences between two groups, a series of statistical tests was employed, including Welch’s test and one-way ANOVA followed by Bonferroni correction. A risk rate of less than 5% (*p* < 0.05) was deemed to represent a statistically significant difference. Statistical significance was set at * *p* < 0.05, ** *p* < 0.01, and *** *p* < 0.001.

## 5. Conclusions

In this study, we demonstrated a primary culture system for AxD astrocytes that can reproduce RFs formation in a cell-autonomous manner. However, functional and genetic abnormalities, such as AxCa and AxGen, were not reproduced. This culture system can be used to elucidate the mechanisms of RFs formation; however, caution should be exercised when elucidating AxD pathology using abnormal function and gene expression of AxD astrocytes as indicators.

## Figures and Tables

**Figure 1 ijms-25-12100-f001:**
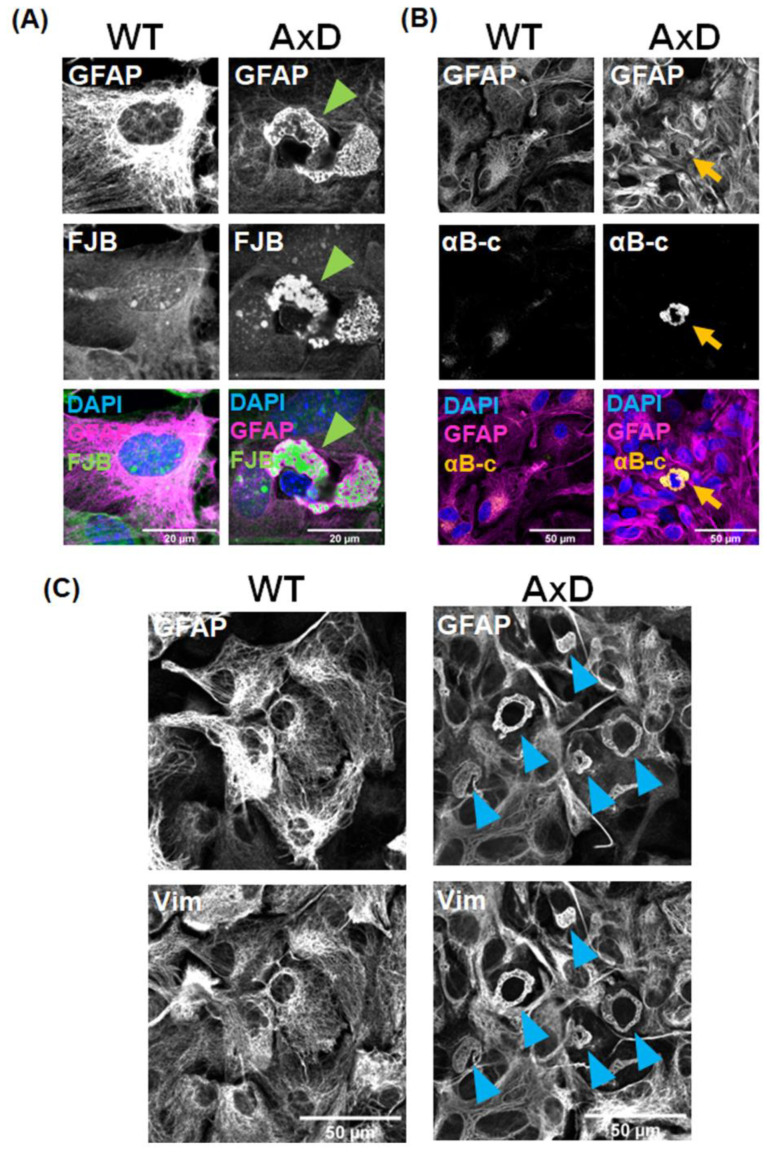
RFs formation in primary cultured AxD astrocytes. RFs, a pathological hallmark of AxD, can be detected by FJB staining and by immunoreactivity of GFAP, αB-crystallin (αB-c), and Vimentin (Vim). (**A**) Representative images of DAPI, GFAP, and FJB staining in AxD and wild-type (WT) cells. RFs are shown by green arrowheads. (**B**) Representative images of DAPI, GFAP, and αB-c in AxD and WT cells. RFs are indicated by orange arrows. (**C**) Representative images of GFAP and Vim in AxD and WT cells. RFs are shown by blue arrowheads.

**Figure 2 ijms-25-12100-f002:**
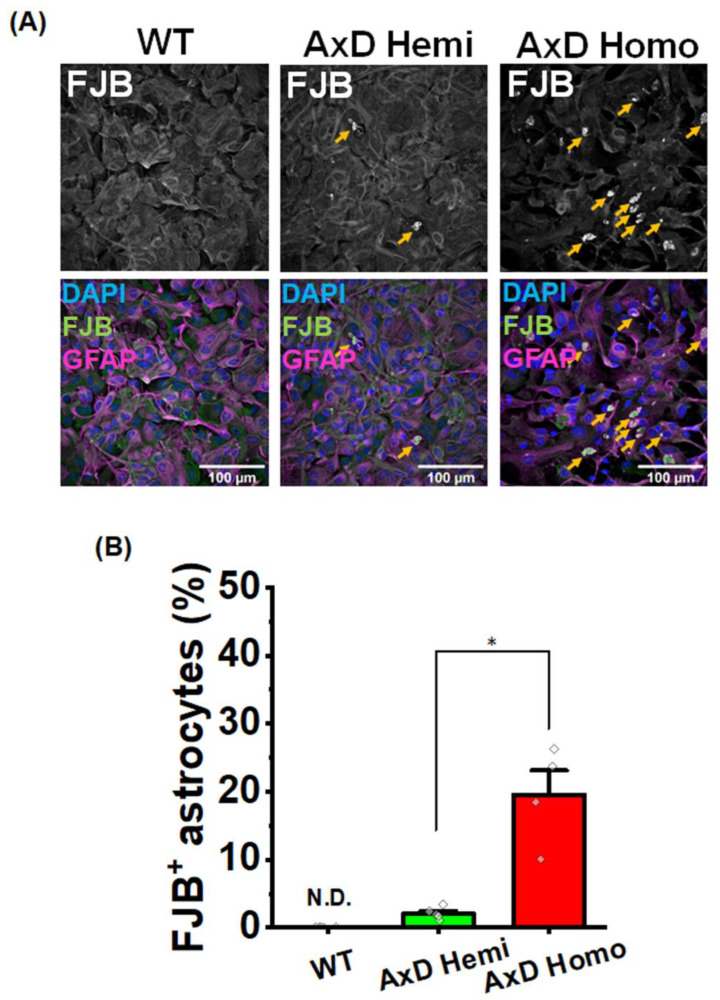
RFs formation depends on the quantity of mutant GFAP. (**A**) Representative images of DAPI, GFAP, and FJB staining in each genotype. RFs are indicated by yellow arrowheads. (**B**) The percentage of FJB^+^/GFAP^+^ cells per a field of view for each genotype. WT: 0%, in 5 field of views from 5 wells; AxD hemizygote (Hemi): 2.05%, in 5 field of views from 5 wells; AxD homozygote (Homo): 19.5%, in 4 field of views from 4 wells. The data are presented as the mean ± SEM. * *p* = 0.016, Welch’s test. N.D.: not detected.

**Figure 3 ijms-25-12100-f003:**
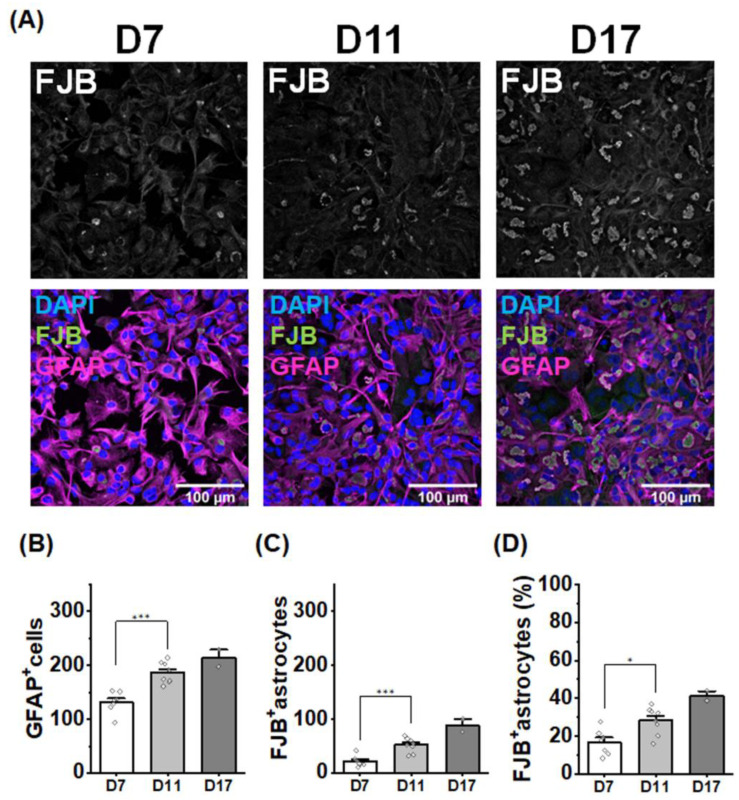
The number of RFs increased with prolonged culture. (**A**) Representative images of DAPI, GFAP, and FJB staining in cells after different incubation times. (**B**) The number of GFAP^+^ cells per a field of view with culture time. D7: in 7 field of views from 7 wells: D11: in 8 field of views from 8 wells: D17: in 2 field of views from 2 wells. The data are presented as the mean ± SEM. D7 vs. D11: *** *p* = 3.6 × 10^−4^, one-way AVOVA with Bonferroni correction. (**C**) The number of FJB^+^ astrocytes per a field of view with culture time. D7: in 7 field of views from 7 wells; D11: in 8 field of views from 8 wells; D17: in 2 field of views from 2 wells. The data are presented as the mean ± SEM. D7 vs. D11: *** *p* = 9.4 × 10^−4^, one-way AVOVA with Bonferroni correction. (**D**) Changes in the percentage of FJB^+^/GFAP^+^ cells per field of view with culture time. D7: in 7 field of views from 7 wells, 16.7%; D11: in 8 field of views from 8 wells, 28.2%; D17: in 2 field of views from 2 wells, 41.2%. The data are presented as the mean ± SEM. D7 vs. D11: * *p* = 0.017, one-way ANOVA with Bonferroni correction.

**Figure 4 ijms-25-12100-f004:**
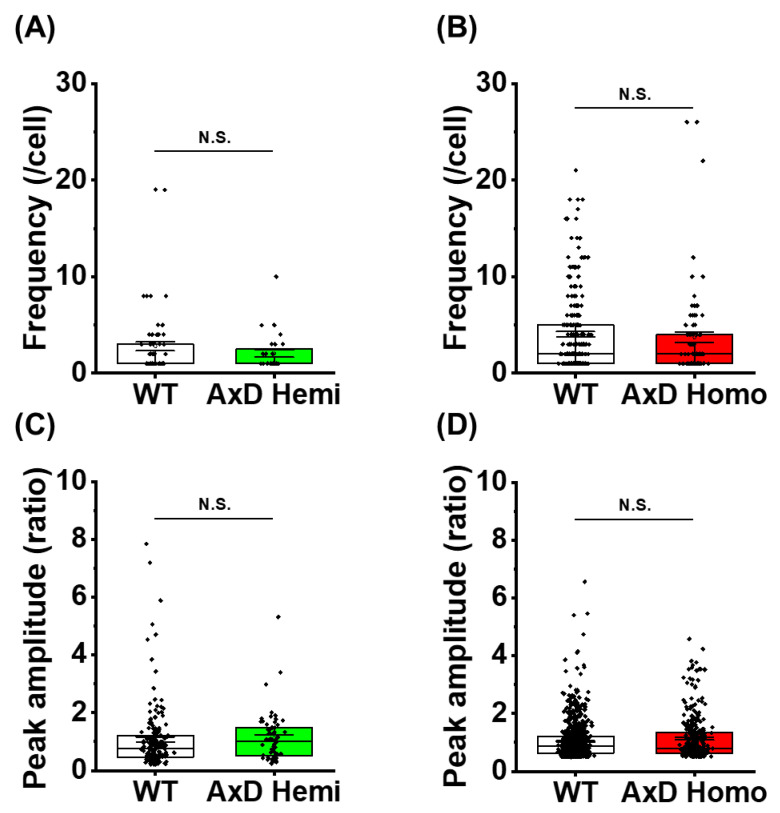
Cultured AxD astrocytes did not exhibit AxCa. (**A**) Frequency of spontaneous Ca^2+^ signals per 10 min in each genotype. WT: 60 cells were analyzed; AxD hemizygote (Hemi): 28 cells were analyzed. Cells showing at least one spontaneous signal were used for analysis. The data are presented as the mean ± SEM. *p* = 0.20, Welch’s test. N.S.: not significant. (**B**) Frequency of spontaneous Ca^2+^ signals per 10 min in each genotype. WT: 227 cells; AxD homozygote (Homo): 83 cells. The data are presented as the mean ± SEM. *p* = 0.57, Welch test. N.S.: not significant. (**C**) Peak amplitude of spontaneous Ca^2+^ signals per 10 min in each genotype. WT: 168 cells; AxD Hemi: 58 cells. Cells showing at least one spontaneous signal were used for analysis. The data are presented as the mean ± SEM. *p* = 0.69, Welch’s test. N.S.: not significant. (**D**) Peak amplitude of spontaneous Ca^2+^ signals per 10 min in each genotype. N.S: not significant.

**Figure 5 ijms-25-12100-f005:**
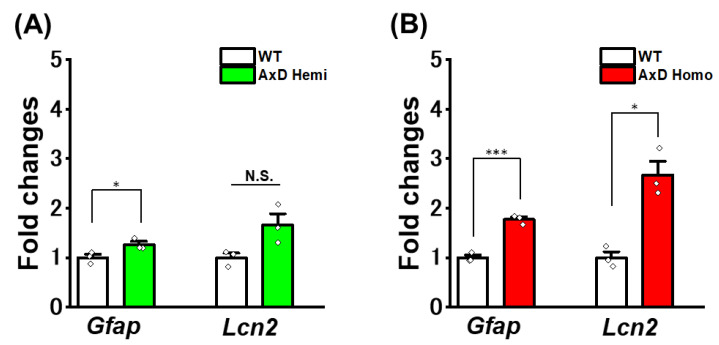
Cultured AxD astrocytes show little AxGen. (**A**) mRNA levels of endogenous *Gfap* and *Lcn2* in each genotype. WT: 3 culture wells were analyzed; AxD homozygote (Homo): 3 culture wells were analyzed. The data are presented as the mean ± SEM. *Gfap*: * *p* = 0.049, Welch’s test. *Lcn2*: *p* = 0.082, Welch’s test. N.S.: not significant. (**B**) mRNA levels of endogenous *Gfap* and *Lcn2* in each genotype. WT: 3 culture wells were analyzed; AxD Homo: 3 culture wells were analyzed. The data are presented as the mean ± SEM. *Gfap*: *** *p* = 5.2 × 10^−4^, Welch’s test. *Lcn2*: * *p* = 0.015, Welch’s test.

**Table 1 ijms-25-12100-t001:** Reagents used.

Reagent	Cat. Number	Company
Poly-L-Lysine (PLL)	P9155	Sigma (St. Louis, MO, USA)
Phosphate Buffered Saline (−) (PBS (−))	14190-144	Gibco (Waltham, MA, USA)
Dulbecco’s Modified Eagle Medium (DMEM)	10313-021	Gibco (Waltham, MA, USA)
Horse Serum (HS)	16050-122	Gibco (Waltham, MA, USA)
Fetal Bovine Serum (FBS)	12483-020	Gibco (Waltham, MA, USA)
Penicillin-Streptomycin	15140-122	Gibco (Waltham, MA, USA)
0.5% trypsin-EDTA	15400-054	Gibco (Waltham, MA, USA)
DNase I	11284932001	Roche (Basel, Switzerland)
Hank’s Balanced Salt Solution (HBSS)	14175-095	Gibco (Waltham, MA, USA)
Fura-2 AM	F1201	Invitrogen (Waltham, MA, USA)
Normal Goat Serum (NGS)	S1000	Funakoshi (Tokyo, Japan)
Triton-X	X100	Sigma (St. Louis, MO, USA)
Paraformaldehyde (PFA)	162-16065	Wako/FUJIFILM (Osaka, Japan)
DMEM/F12	10565-018	Gibco (Waltham, MA, USA)
B27 supplement	17504-044	Gibco (Waltham, MA, USA)
70 µm cell strainer	352350	Corning (Corning, NY, USA)
EMEM	30-2003	ATCC (Manassas, VA, USA)
GlutaMax	35050-061	Gibco (Waltham, MA, USA)
Fluoro-Jade B	AG310	Millipore (Burlington, MA, USA)
Prime Time Gene Expression MasterMix	1055771	IDT (Coralville, IA, USA)
PrimeScript RT Reagent Kit	RR037A	Takara (Shiga, Japan)
4′,6-diamidino-2-phenylindole (DAPI)	PP089	DOJINDO (Kumamoto, Japan)

**Table 2 ijms-25-12100-t002:** IDT probe used.

Target Gene	Cat. Number	Sequence
*Gapdh*	Mm.PT.39a.1	Forward; 5′-GTGGAGTCATACTGGAACATGTAG-3′ Reverse; 5-AATGGTGAAGGTCGGTGTG-3
*Gfap*	Mm.PT.58.31297710	Forward; 5′-AACCGCATCACCATTCCTG-3′ Reverse; 5-GCATCTCCACAGTCTTTACCA-3
*Lcn2*	Mm.PT.58.10167155	Forward; 5′-CCTGTGCATATTTCCCAGAGT-3′ Reverse; 5-CTACAATGTCACCTCCATCCTG-3

**Table 3 ijms-25-12100-t003:** Antibody used.

Antibody	Cat. Number	Company	Dilution
Rat IgG anti-GFAP antibody	13-0300	Invitrogen (Waltham, MA, USA)	1:1000
Rabbit IgG anti-GFAP antibody	AB5804	Millipore (Burlington, MA, USA)	1:1000
Chicken IgG anti-Vimentin antibody	ab24525	Abcam (Cambridge, UK)	1:500
Mouse IgG anti-αB-crystallin	ab13496	Abcam (Cambridge, UK)	1:250
Goat IgG Alexa fluor 546 anti-rat IgG	a11081	Invitrogen (Waltham, MA, USA)	1:1000
Goat IgG Alexa fluor 488 anti-rabbit IgG	a11034	Invitrogen (Waltham, MA, USA)	1:1000
Goat IgG Alexa fluor 488 anti-mouse IgG	a11029	Invitrogen (Waltham, MA, USA)	1:1000
Goat IgG Alexa fluoro 546 anti-rabbit IgG	a11035	Invitrogen (Waltham, MA, USA)	1:1000

## Data Availability

The data that support the findings of this study are available from the corresponding author upon reasonable request.

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
