# Peer review of "Establishment and Use of Primary Cultured Astrocytes from Alexander Disease Model Mice"

_ijms, 2024, doi:10.3390/ijms252212100_

Round 1
Reviewer 1 Report
Comments and Suggestions for Authors
This study by Kubota et al. establishes and utilizes primary cultured astrocytes derived from Alexander disease (AxD) model mice to investigate the cell-autonomous and non-cell-autonomous mechanisms involved in AxD pathogenesis. The authors focus on three key features of AxD astrocytes: Rosenthal fibers (RFs), aberrant calcium signaling (AxCa), and upregulation of disease-associated genes (AxGen). The study demonstrates that RF formation in AxD astrocytes is driven by a cell-autonomous mechanism, while AxCa and AxGen are likely non-cell-autonomous, requiring interactions with other cells such as neurons or microglia. These findings highlight the utility of primary AxD astrocyte cultures for studying RF formation but indicate limitations in their ability to model AxCa and AxGen.
Main Concern:
- Interpretation of non-cell-autonomous nature of AxCa and AxGen: A major concern is the interpretation of the non-cell-autonomous nature of AxCa and AxGen. The study suggests that these features depend on interactions with other cell types (e.g., neurons or microglia). However, it is possible that the negative results for AxCa and AxGen are primarily due to the fact that astrocytes in vitro do not reach full maturity and tend to enter a reactive state (characterized by upregulation of GFAP). This may mask or alter the true nature of these features, rather than indicating an inherent reliance on input from other cell types. Maybe the co-culture with WT neurons could ameliorate their maturation.
- Heterogeneity astrocyte: Another concern is the choice to use astrocytes from both the cortex and hippocampus. Astrocytes from these two regions are known to exhibit profound molecular differenceis differences, including distinct GFAP expression levels, which could introduce additional heterogeneity into the analysis. It would be important to clarify why this choice was made and whether a more region-specific dissection (focusing only on the cortex, for instance) would yield more consistent or interpretable results.
Minor Concerns:
- Figure 5: The authors should provide images to support the analysis shown in Figure 5, allowing for qualitative evaluation alongside the quantitative data. Additionally, the analysis of genes that were not differentially expressed between the two conditions should also be included, even if the results are negative, to provide a more comprehensive understanding of the dataset.
- Atp2a2 data: The authors mention that Atp2a2 was not downregulated, but the supporting data should be included in the manuscript to validate this observation.
- Calcium Imaging: The specific time point at which calcium imaging was performed should be clearly stated in the main text. Additionally, would keeping the astrocytes in culture for a longer period improve the maturation of the cells and, in turn, enhance the reliability of the calcium imaging results?
- Sample size in figure 5: Increasing the number of biological replicates (currently n=3) could improve the statistical power of the analysis. Some results that are currently not significant may become significant with a larger sample size, providing more robust conclusions.
Reviewer 2 Report
Comments and Suggestions for Authors
Good manuscript that clearly presents obtained results and their explanation. The advantage of the recent work is that results are compared with previously published data obtained in vivo that allows authors to get interesting comparison between in vivo and in vitro.
Despite overall positive impression the manuscript does not avoid some negative points that should be fixed to create really good paper.
Thus line 91 –to delete “their interior” that completely not clear what does it means.
Figure 1. some astrocytes are stained for vimentin but not for GFAP ( in WT and AxD cultures). Better to give short remark on this point.
l. 123 – better to change “FJB+ cells” for “FJB+ astrocytes” to avoid misunderstanding.
l.119 – should be “increases”
l.120 – “development’ better to change for ”age”
Figure 3 (C) FJB+ cells recommended to change for “FJB+ astrocytes” to avoid misunderstanding.
L.187 – to add astrocyte between primary and cultures.
l.190 “depends on the developmental process” -quite unclear statement that does give any explanation only produces not necessary questions – delete.
l.189-190 statement that “accumulation depends on the developmental process” does not have any sense without clear explanation
l.202 “stained in vivo”….”stained RFs in vitro.” Should be changed to avoid misunderstanding – staining cannot be done in vivo or in vitro
